# Research on Dust Effect for MEMS Thermal Wind Sensors

**DOI:** 10.3390/s23125533

**Published:** 2023-06-13

**Authors:** Zhenxiang Yi, Yishan Wang, Ming Qin, Qingan Huang

**Affiliations:** Key Laboratory of MEMS of the Ministry of Education, Southeast University, Nanjing 210096, China

**Keywords:** thermal wind sensor, dust effect, temperature difference, accumulated dust

## Abstract

This communication investigated the dust effect on microelectromechanical system (MEMS) thermal wind sensors, with an aim to evaluate performance in practical applications. An equivalent circuit was established to analyze the temperature gradient influenced by dust accumulation on the sensor’s surface. The finite element method (FEM) simulation was carried out to verify the proposed model using COMSOL Multiphysics software. In experiments, dust was accumulated on the sensor’s surface by two different methods. The measured results indicated that the output voltage for the sensor with dust on its surface was a little smaller than that of the sensor without dust at the same wind speed, which can degrade the measurement sensitivity and accuracy. Compared to the sensor without dust, the average voltage was reduced by about 1.91% and 3.75% when the dustiness was 0.04 g/mL and 0.12 g/mL, respectively. The results can provide a reference for the actual application of thermal wind sensors in harsh environments.

## 1. Introduction

Thermal wind sensors based on the MEMS technology have been developed for many years since it was first proposed in 1990 [1]. Consisting of heating resistors and temperature sensors on the chip, the thermal wind sensor could achieve a full 360° sensitivity by measuring the temperature gradient in two directions perpendicular to each other [1,2,3]. It has had many advantages such as small size, fast response, low cost and low power consumption nowadays and is widely used in various fields such as environment monitoring and agriculture applications [3,4,5,6]. Up to now, many efforts have been made to enhance the performance of the MEMS wind sensors. First, in chip fabrication, the substrate is always thinned and trenches are also applied between the sensing elements [7,8]. Those designs can reduce power consumption by suppressing unwanted lateral heat conduction in the substrate. In terms of packaging, the chip is always filled with glue and the backside is used as a sensing surface [9,10]. This can protect the devices to a certain extent and guarantee the robustness of the sensor.

The great potential of MEMS-based sensors has been demonstrated by improved performance of sensors in harsh environments [11,12,13]. MEMS technologies allow access to a broad range of unexpected new applications opportunities. Thermal wind sensors experience many complicated working conditions in practice. Above all, the effects of the environment influence the accuracy of the MEMS thermal wind sensors, and our group has previously researched the temperature and humidity effects [14,15,16]. 

However, in application, it has been found that dust will always accumulate on the surface and may influence the sensor’s performance, leading to a degraded sensitivity and measurement accuracy. To our knowledge, so far, there has been little research conducted on the dust effect in the literature. Consequently, for the first time, this paper investigates the influence of the dust accumulated on the surface. An equivalent circuit model was established to explain the dust effect theoretically. In order to verify the proposed model, a FEM simulation has been performed and the sensor with dust on its surface was tested in the wind tunnel. Finally, related results and discussion have been presented.

## 2. Theory and Simulation

The MEMS thermal wind sensors consist of a central heater and four thermistors symmetrically arranged around in two perpendicular directions. The spatially separated temperature sensing elements can detect the temperature distribution on the sensor’s surface. Figure 1a illustrates the working principle of the traditional sensor using back-surface sensing, while Figure 1b shows the heat transfer principle for the sensor with dust accumulated on its surface.

When the wind passes over the sensor, the heat transfer from the surface to the flow would be larger on the upstream side than on the downstream side, making the temperature downstream (*T*_2_) a little higher than upstream (*T*_1_). The measured temperature difference *δT = T*_2_ − *T*_1_ can be obtained as follows [17]:(1)$$\delta T=c\cdot k_{s}{}^{-1}\cdot k_{f}^{2/3}\mu^{-1/6}\rho^{1/2}C_{p}^{1/3}\cdot U^{1/2}\cdot \Delta T$$
where *c* is a constant related to the size of the sensor and *k_s_* refers to the thermal conductivity of the substrate. The variables *k_f_*, *ρ*, *C_p_* and *μ* refer to the thermal conductivity, density, heat capacity and dynamic viscosity of the wind, respectively, while *U* refers to the wind speed. Obviously, *δT* increases with the flow velocity, so it can be used to determine the magnitude of the wind [18].

Usually dust accumulated on the surface is not compact and has a relatively small thickness. Therefore, it influences the performance of the sensor mainly by changing the solid heat transfer in the vertical direction as shown in Figure 1b.

The equivalent circuit models of the sensor and the sensor with dust on its surface are displayed in Figure 2a and Figure 2b, respectively. We use *R* to represent the resistance against heat transfer during operation. In the circuits, *R*_1_ and *R*_2_ represent the thermal resistance of the plastic-sealing substrate and the blackened aluminum ring, respectively, while *R_f_* refers to the thermal convection resistance caused by the fluid flowing over the sensor’s surface. Ignoring the dust in-layer lateral heat conduction, the surface-attached dust may increase the vertical heat conduction in operation. As shown in the equivalent circuit in Figure 2b, a conductive thermal resistance *R_dust_* is connected with the convective thermal resistance in series. In simplifying the accumulated dust as a single-layer plane wall, the steady heat conduction thermal resistance is denoted as [19]
(2)$$R_{dust}=\delta /(k_{dust}A)$$
where *δ*, *k_dust_* and *A* refer to the thickness, thermal conductivity and cross-sectional area of the dust, respectively.

With dust accumulating on the surface, the temperature actually used for thermal convection changes from *T_chip_* to *T*′, which will decrease the temperature difference that causes the convective heat transfer between the chip surface and the fluid. It can be seen that the dust accumulation on the sensor’s surface can reduce the temperature difference in two perpendicular directions. Further, for the same wind speed, the more the dust accumulates on the surface, the smaller the temperature difference is.

To verify the analysis above, a FEM simulation was carried out by using COMSOL Multiphysics software. We established a simulation model using its back surface for sensing. The thermal wind sensor is fixed in a plastic-sealing substrate, surrounded by a blackened aluminum ring to realize the heat sink. With all other things being equal, the dust effect would be mainly reflected with decreasing the temperature for thermal convection only when the vertical in-layer thermal conduction is considered. Figure 3a illustrates that the temperature on the sensing surface decreases with dust accumulation, further influencing the convective heat transfer in operation. The intuitive influence of the accumulated dust is presented in the thermal field distribution of the sensing surface shown in Figure 3b.

As mentioned above, the temperature difference between the upstream and downstream sides along one direction can be used to determine the wind velocity. The *δT* under the same conditions therefore reflects the sensitivity of thermal wind sensors. Figure 4a shows the average temperature difference of two thermistors varying with wind speed up to 20 m/s with dust thickness of 0 μm, 100 µm and 200 µm, respectively. Figure 4b shows the simulated temperature difference as a function of dust amount, in which the wind speed is kept at 10 m/s, while the thickness of the dust is changed from 20 µm to 200 µm. Evidently, the temperature difference reduces when the dust amount increases. This indicates that the accumulated dust has a negative influence on the performance of the sensor, which is consistent with the theoretical analysis.

Figure 5 shows the influence of dust thermal conductivity on the temperature difference between the upstream and downstream sides. When the thermal conductivity increases from 0.172 W/(m·K) to 0.372 W/(m·K), the temperature difference will rise when the wind speed is constant.

Moreover, the situation in which the dust drops on the sensor’s surface irregularly was also considered and the simulation results are presented in Figure 6. Clearly, the temperature difference is reduced when the dust is located upstream for a fixed wind speed.

## 3. Experiments and Discussions

To realize the dust accumulation on the surface, two methods were designed by choosing quartz sand instead of dust. For the first method, fixed quantitative dust is put into a box and then the wind sensor with a moist surface is put into the box. After that, the fan is turned on in the sealed box to blow the dust up inside the box. Finally, the fan is turned off and kept for a while to realize the dust’s deposition on the sensor’s surface. This method quantifies the surface dust by controlling the dust weight within the fixed volume; that is, the dust concentration inside the box environment (in kg/m^3^). Another approach is that dust-suspension liquid with different concentration is spin-coated on the sensor’s surface and then dried. In the experiment, the total amount of suspension liquid for each time and the relative position between the spray gun and the sensor remained unchanged. Figure 7 displays the photograph of the sensor with different dust accumulation on its surface.

Measurements were performed in the wind tunnel where wind speed could be controlled by adjusting its working frequency. After installing the sensor without dust, and with a different amount of dust in the wind tunnel in turn, the output voltage is measured as MCU with Vx and Vy changing with wind speed. The measurement condition is that the temperature is about 20 °C and the humidity is close to 60% RH.

Figure 8 depicts the measured output voltage as a function of wind speed from 0 m/s to 19.1 m/s. An abnormal point exists in Figure 8a at 5 m/s and it may be caused by the increasing wind speed. The dust accumulated on the surface was blown away so that the practical amount of the dust was reduced when the wind speed increased. Before this point, the measured voltages with and without dust were compared. The results demonstrated that the sensor’s output voltage was reduced by 5% and even 24% at the same wind speed when the dustiness was 1.28 kg/m^3^ and 6.4 kg/m^3^, respectively. The sensor with dust on the surface using the second method was tested in the wind tunnel as well. As shown in Figure 8b, the measured curve was smooth and there was no obvious irregular point. This was mainly because the dust on the sensor’s surface was comparatively stable. The experimental results indicated that, compared to the sensor without dust, the output voltage was reduced by about 1.91% and 3.75% when the dustiness was 0.04 g/mL and 0.12 g/mL, respectively.

We did not take the dust slip during wind flow into consideration in the simulation and the dust layer was assumed to be fixed on the surface. However, the model idealized the situations. The first method mentioned above is closer to the real-world situation, where the dust amount will reduce as the wind speed increases. The second method focuses on investigating the influence of the accumulated dust amount on the sensor performance, where the dust is relatively fixed. The experiment results obtained by different methods both demonstrate that the measured output voltage would decrease by the amount of dust attached to the sensor’s surface. This means that the accumulated dust can reduce the temperature difference between thermistors on upstream and downstream sides and therefore degrade the sensitivity of the wind sensor. In a way, the experimental results were accordant with the theoretical analysis and simulation, which verified the negative effect of the surface dust on the sensor’s performance. Further, the dust will have an effect on the output voltage reduction for the MEMS wind sensors. In addition, the dust amount on the surface will vary with the wind speed in practical conditions and will therefore reduce the measurement accuracy as well.

## 4. Conclusions

This communication investigated the effect of dust accumulation on the surface of the MEMS thermal wind sensors. The equivalent circuit model was established for theoretical analysis, and a FEM simulation was carried out by using COMSOL Multiphysics software. Two different methods were utilized to realize the adhesion of dust to the sensor’s surface. The experimental results indicated that the dust would degrade the sensor performance, including sensitivity and accuracy. Additionally, the sensor may deteriorate further with increased amounts of dust. The proposed model can be extended to other harsh environments in which sensors may employed.

## Figures and Tables

**Figure 1 sensors-23-05533-f001:**
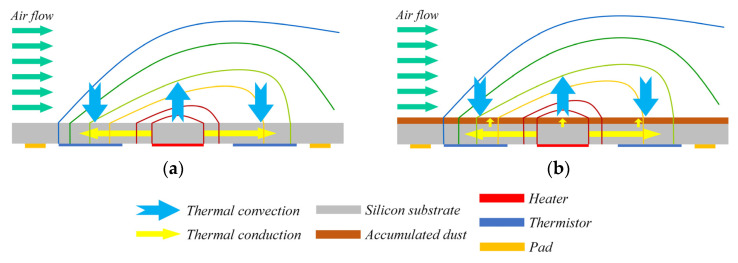
Working principle of wind sensor (**a**) without dust and (**b**) with dust on the back surface.

**Figure 2 sensors-23-05533-f002:**
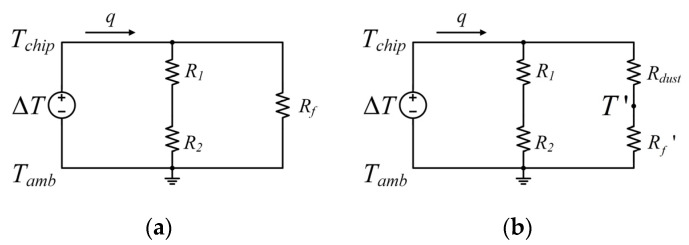
The equivalent circuit models: (**a**) the thermal wind sensor and (**b**) the thermal wind sensor with accumulated dust on its surface.

**Figure 3 sensors-23-05533-f003:**
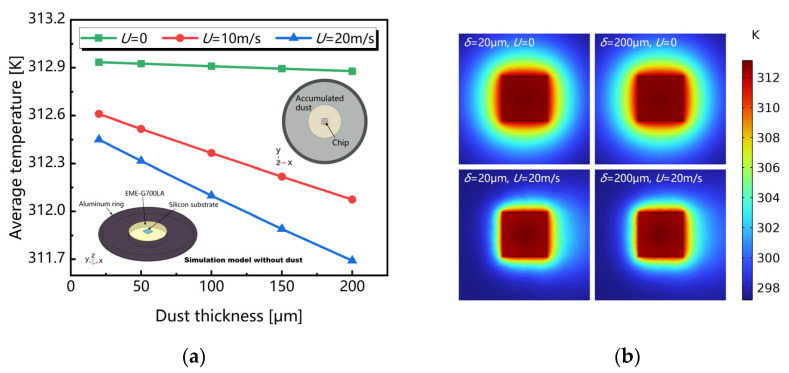
Simulated results: (**a**) average temperature and (**b**) the thermal field distribution of the sensor’s surface varying with dust thickness under different wind speeds.

**Figure 4 sensors-23-05533-f004:**
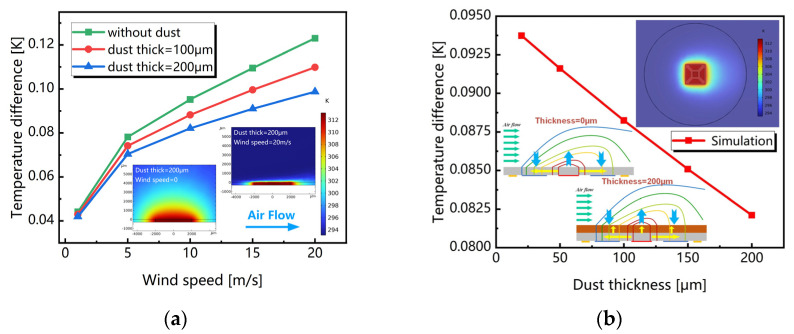
Simulated temperature differences: (**a**) changing with wind speed and (**b**) changing with dust thickness.

**Figure 5 sensors-23-05533-f005:**
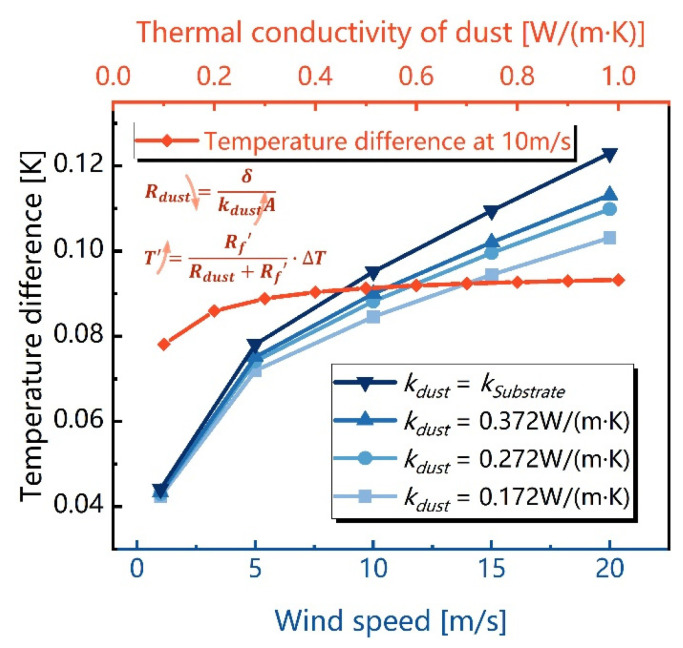
Simulated temperature differences influenced by the thermal conductivity of dust.

**Figure 6 sensors-23-05533-f006:**
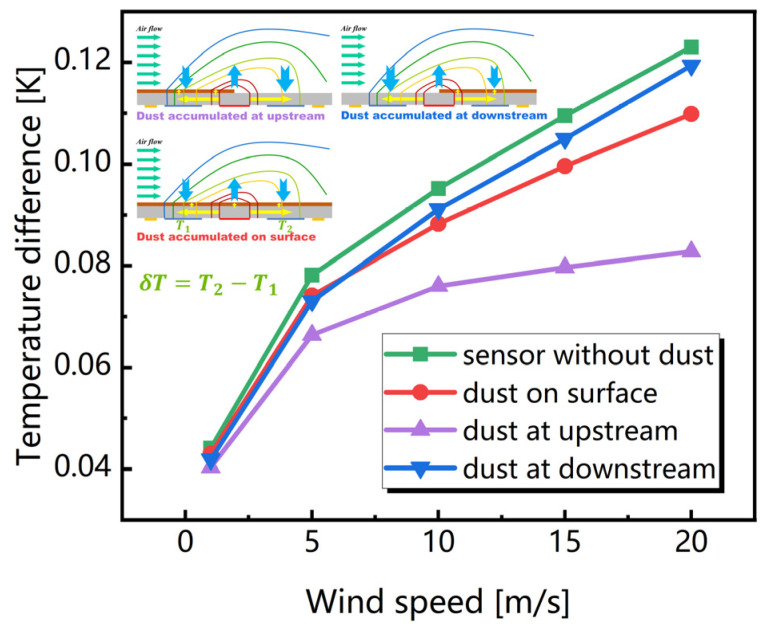
Simulated temperature differences caused by the unequal distributions of dust.

**Figure 7 sensors-23-05533-f007:**
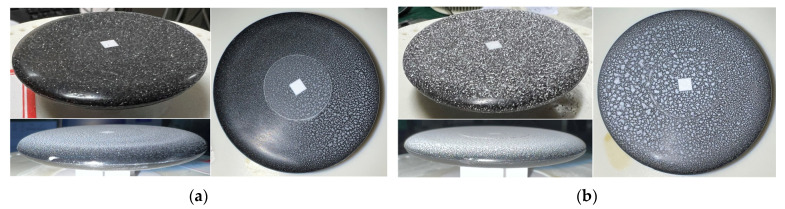
Photographs of dust adhesion to the sensor’s surface: (**a**) with less dust and (**b**) with more dust.

**Figure 8 sensors-23-05533-f008:**
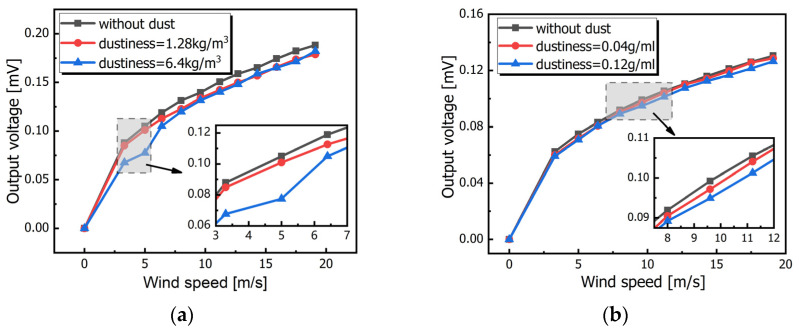
Measured voltage of the sensor under different dust environments: (**a**) first method and (**b**) second method.

## Data Availability

Not applicable.

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
