# Peer review of "Research on Dust Effect for MEMS Thermal Wind Sensors"

_sensors, 2023, doi:10.3390/s23125533_

Round 1

Reviewer 1 Report

This manuscript reports MEMS wind sensor based on dust effect. The voltage can be reduced 1.91% and 3.75% with dusting 0.04g/ml and 0.12g/ml. The manuscript can be accpeted after the following issues are solved.

1, The backgroud is not solid enough, many sensors relating materials did not explained well. Also, there are only 18 references be cited, the authors can cite more reference to support the sensors, such as Development trends and perspectives of future sensors and MEMS/NEMS. Micromachines. 11, 7, 541, 2020.

2, What is the real purpose of Figure 1 with and without dust? Any background on the dust? Any theory to support the theory?

3, How to demonstrate the equivalent cuicuit of Figure 2?

4, As mention the thickness of the dust, how about the temperature and the humidity? How about the wind direction on the effect of the device? As known the temperature difference, how about the temperature itself?

5, How about the weight of the dust to the results?

6, How about the wind blow away the dust? How to identify this case?

The english is fine to understand

Reviewer 2 Report

The paper is interesting and timely.

The topic of noisy sensors is an appropriate topic today.

The authors discuss an innovative method to evaluate the imperfections.

It is well done.

Moreover the imperfection could hel the sensors to work better.

There are significative examples in the literature.

Therefore I suggest to remark some items in this aspect and to cite the following relevant contribution:

EEE AccessOpen AccessVolume 9, Pages 29573 - 295832021 Article number 9351920 Document type Article• Gold Open Access Source type Journal ISSN 21693536 DOI 10.1109/ACCESS.2021.3058506 View more   

Imperfections in Integrated Devices Allow the Emergence of Unexpected Strange Attractors in Electronic Circuits

  • Bucolo, Maideab;
  • Buscarino, ArturoabSend mail to Buscarino A.;
  • Famoso, Carloa;
  • Fortuna, Luigiab;
  • Gagliano, Salvinaa

Review the english.

Reviewer 3 Report

The manuscript presents theoretical and experimental studies on dust accumulation influences over the MEMS performances. In principle this study is interesting and potentially useful for some readers. However, there are few things which in my opinion could be improved.

Experimental:

Authors are considering ‘dust’ influence using quartz sand with different thicknesses. I am not sure how relevant is for the potential users this material, and I would suggest considering the change of the “dust composition” by considering variations of the “thermal capacity” of the “dust”. In other words, to try to evaluate the change of the results by potential changes of the thermal capacity of the “dust”. This could mean measurements for few more “dust compositions” but could also mean at least to estimate the influence of the thermal coefficient changes.

Furthermore, the authors could also try to estimate (at least in principle) the influence of an unequal distribution (over two opposite thermistors) of the dust (something like error bars for specific non-uniformity)

Presentation:

The authors are presenting separate theoretical and experimental results on the dust accumulation over the MEMS sensor. My suggestion would be to try to present on a graph both results as normalized results for the “theoretical” and “experimental” variation errors due to the accumulated dust, over the “measured wind”, in order to have a ‘visual’ confirmation of the correctness of the presented simulations.

Reviewer 4 Report

This paper proposes Research on Dust Effect for MEMS Thermal Wind Sensors. The paper is well written, but in my opinion, it lacks a better description of the state of the art and a better explanation of the novelty of this work. Especially, the authors should compare their linearity and resolution results with the literature in order to demonstrate their contribution.

page 2 line 58: where c is a constant related to the size of the sensor, ks refers to the thermal conductivity of the substrate, kf, ρ, Cp and μ are the thermal conductivity, density, heat capacity at constant pressure and dynamic viscosity of the fluid ? The sentence is unclear.

Why are the image Figure 3 and  4 are different? in figure show the thermistor.

Improve the figure 3 

Details are lacking on how the voltage measurement will be carried out, model, and the measurement conditions. 

the resolution scale is short, about two degrees, what is the margin of error?

Improve figure 5,  it's not clear what the want to show

  •  

Minor editing of English language required

Round 2

Reviewer 1 Report

The authors did answer all my issues. I recomment to publish in the current form.

Author Response

Thanks a lot for the reviewer.

Reviewer 4 Report

Dear Authors 

The manuscript looks better.

Author Response

Thanks a lot for the reviewer.